# Poly[(*N*-acryloyl glycinamide)-*co*-(*N*-acryloyl l-alaninamide)] and Their Ability to Form Thermo-Responsive Hydrogels for Sustained Drug Delivery

**DOI:** 10.3390/gels5010013

**Published:** 2019-03-03

**Authors:** Mahfoud Boustta, Michel Vert

**Affiliations:** Department of Artificial Biopolymer, Institute for Biomolecules Max Mousseron, UMR CNRS 5247, Faculty of Pharmacy, University of Montpellier-CNRS-ENSCM, 15 Avenue Charles Flahault, BP 14491, 34093 Montpellier CEDEX 5, France; mahfoud.boustta@umontpellier.fr

**Keywords:** *N*-acryloyl glycinamide, *N*-acryloyl glycinamide (NAGA), *N*-acryloyl l-alaninamide, NAGA-*co*-*N*-acryloyl l-alaninamide chiral analog (NAlALA) copolymers, drug delivery, gel↔sol transition, Upper Critical Solution Temperature (UCST), thermo-responsive hydrogel

## Abstract

In the presence of water, poly(*N*-acryloyl glycinamide) homopolymers form highly swollen hydrogels that undergo fast and reversible gel↔sol transitions on heating. According to the literature, the transition temperature depends on concentration and average molecular weight, and in the case of copolymers, composition and hydrophilic/hydrophobic character. In this article, we wish to introduce new copolymers made by free radical polymerization of mixtures of *N*-acryloyl glycinamide and of its analog optically active *N*-acryloyl l-alaninamide in various proportions. The *N*-acryloyl l-alaninamide monomer was selected in attempts to introduce hydrophobicity and chirality in addition to thermo-responsiveness of the Upper Critical Solubilization Temperature-type. The characterization of the resulting copolymers included solubility in solvents, dynamic viscosity in solution, Fourrier Transform Infrared, Nuclear Magnetic Resonance, and Circular Dichroism spectra. Gel→sol transition temperatures were determined in phosphate buffer (pH = 7.4, isotonic to 320 mOsm/dm^3^). The release characteristics of hydrophilic Methylene Blue and hydrophobic Risperidone entrapped in poly(*N*-acryloyl glycinamide) and in two copolymers containing 50 and 75% of alanine-based units, respectively, were compared. It was found that increasing the content in *N*-acryloyl-alaninamide-based units increased the gel→sol transition temperature, decreased the gel consistency, and increased the release rate of Risperidone, but not that of Methylene Blue, with respect to homo poly(*N*-acryloyl glycinamide). The increase observed in the case of Risperidone appeared to be related to the hydrophobicity generated by alanine residues.

## 1. Introduction

Polymeric hydrogels are swollen 3D matrices composed of at least 50% or more of aqueous medium. Two domains of applications of hydrogels are drug delivery and tissue engineering [1,2,3,4,5,6]. In such hydrogels, water is entrapped in a 3D network of chemically or physically cross-linked macromolecules. Contrary to chemical cross-linking, physical cross-linking can be reversible under the action of a change in environmental conditions. The change is referred to as a stimulus when the action is desired. Basically, the reversibility of the physical cross-linking present in hydrogels results in gel↔sol transition. Stimuli like salt concentration, pH, ionic strength, and temperature changes have been proposed as a means to modify water-polymer interactions and lead to sol↔gel reversible transitions [5]. Many injectable hydrogels based on artificial polymers or on biopolymers have been studied over the past decades for their interest in various domains of therapy [7,8]. In mammalians, ionic strength, pH, and temperature are set at physiological values that may serve as references to make an in situ sol→gel transition respectful of living elements (proteins, cells, etc.) and exploitable in therapy. For instance, a stimulus-responsive gel-forming system can be injected in the sol form slightly above or below such a physiological reference, and in gel in situ soon after being under physiological conditions. With respect to this, temperature has been considered particularly attractive. Among the thermo-responsive polymer-water systems, a majority is based on the Low Critical Solution Temperature (LCST) phenomenon, i.e., they are solutions at temperatures lower than body temperature [9,10]. LCST-based transitions are generally due to dehydration phenomena, and thus results in water expulsion and matrix contraction. Although less frequently, thermo-responsive hydrogels based on the Upper Critical Solution Temperature (UCST) phenomenon are also present in the literature [9,11]. When sol is above the transition temperature, the aqueous medium turns to a gel on cooling. In this case, gelation is generally due to the formation of a network of hydrogen bonds with almost no contraction. Gelatin is the archetype of such hydrogels. However, gelatin is rapidly degraded in vivo and has a sol→gel transition temperature around 32 °C that is too low relative to body temperature.

It is now well known that some poly(*N*-acryloyl glycinamide) polymers (PNAGA)-water systems exhibit reversible sol↔gel transitions of the UCST type [12]. In such systems, the polymer is soluble above the UCST, whereas more or less consistent hydrogels are rapidly formed below this [13,14,15,16]. Polymer concentration, molecular weights, and copolymerization are factors that can be exploited to modify the temperature of phase separation or gelation. *N*-acryloyl glycinamide (NAGA) was copolymerized with different co-monomers, namely *N*-acetyl acrylamide, acrylic acid, butyl acrylate, styrene, biotin-functionalized methacrylamide, etc. [12,17,18,19]. When occurring, the UCST-dependent phase separation was affected by the presence of the co-monomers and even suppressed when carboxylic functions were present [12,18]. 

In a previous publication, we have shown that drug-containing PNAGA hydrogels present an interesting potential to fulfill many of the properties required for drug delivery after injection under warm sol form [20,21]. The potential was investigated using Methylene Blue as a model of hydrophilic drugs [21]. In vitro, the release of this dye was diffusion-controlled and sustained release was demonstrated in vivo. In contrast, Risperidone, a hydrophobic drug, loaded in a PNAGA hydrogel exhibited solubility-controlled zero order release [21], the release rate being slowed down when Mg(OH)_2_ was present in the formulation [22]. 

This article aims at exploring a new type of optically active and thermo-responsive polymer obtained by copolymerization of NAGA and its *N*-acryloyl l-alaninamide chiral analog (NAlALA) (Figure 1). 

Based on the protocol exploited for the free-radical polymerization of NAGA, copolymers combining NAGA and NAlALA-derived units were synthesized using isopropanol as transfer agent to control and adjust molecular weights. The characterization of homo and copolymers made from feeds containing 0, 10, 25, 50, 75, and 100% NAlALA included solubility in water and solvents, Fourier Transform Infrared (FTIR) and Nuclear Magnetic Resonance (NMR) spectra, dynamic viscosity, chiroptical properties, and gel→sol transition temperature. Methylene Blue and Risperidone, used previously to test the potential of PNAGA hydrogels, were individually combined with the copolymers derived from approximately 50:50 and 25:75 M:M NAGA:NAlALA polymerization feeds. Release profiles were discussed comparatively. All data were used to discuss the influence of the presence of NAlALA methyl groups on the properties of the copolymers relative to homopolymers.

## 2. Results and Discussion

### 2.1. NAGA and NAlALA Monomers

NAlALA was synthesized using a protocol similar to that reported for NAGA [20,21] modified from [13]. Both compounds were soluble in aqueous media. Figure 2 compares the ^1^H-NMR spectra of NAGA and NAlALA in d6-DMSO, with respective peak assignments. The two spectra are rather comparable in the 8.5–5.5 ppm range. Below 5.5 ppm, the NAlALA methyne **2** resonance (quintuplet at 4.31 ppm with J = 7.28 Hz) is located downfield compared to the NAGA **d** methylene one (doublet at 3.65 ppm). The presence of the quintuplet shows that the protons of the CH–CH_3_ group were coupled with the proton of the vicinal NH amide group. In the NAGA spectrum, the methylene **d** protons and the NH proton were also coupled and appears as doublet and triplet, respectively. Furthermore, the NALALA spectrum presents a doublet at 1.22 ppm specific of methyl **1** (J = 7.11 Hz).

Figure 3 shows the ^13^C-NMR spectrum of NAlALA in d6-DMSO with the peak assignment that agrees with the structure deduced from the ^1^H-NMR spectrum in Figure 2.

### 2.2. Synthesis and Characterization of Poly(NAGA-co-NAlALA) Copolymers

It was previously reported that the molecular weight of PNAGA polymers synthesized by free radical polymerization can be controlled using isopropanol as the transfer agent [15]. The method was applied to mixtures of NAGA and NAlALA and yielded PNAGA-*co*-NAlALA copolymers irrespective of the proportion of the two acrylic monomers in the feed. Compositions in glycinamide and l-alaninamide-derived units deduced from ^1^H-NMR spectra (Figure 4) were close to those of the corresponding feeds after purification by dialysis and freeze drying (Table 1). This finding argued in favor of monomers with similar reactivity.

According to the normal action of transfer agents in chain polymerization of vinylic and acrylic monomers, the higher the concentration in transfer agent, the lower the molecular weight is. Therefore, the molecular weights of the polymers were likely to increase when the transfer agent concentration decreased from 1 to 0.1 M, a trend confirmed by dynamic viscosity data given in Table 1, where abbreviations of copolymers are composed of the acronym COP followed by a number that reflects the percentage of NAlALA-based units in the polymerization medium and a subscript number that reflects the concentration of isopropanol in the polymerization medium.

Table 1 shows that the physical aspect of the different polymers in saline at room temperature depended on the concentration of the polymerization medium in transfer agent for a given concentration in monomers (mol/dm^3^) (comparison of COP50_0.1, 0.25 and 1_) and on the composition in monomers (comparison of COP25_0.25_, COP50_0.25_, and COP75_0.25_) in the aqueous polymerization medium. Gel formation depended also on the polymer concentration. At 1% W/V, none of the copolymers gelled at room temperature except COP50_0.1_, which had the higher molecular weight. At 5%, all the polymers were under gel form. However, the gels were more or less consistent and cohesive, as shown by the ranking from grade 3 (turbid soft gel) to grade 6 (elastic gel). Therefore, it was concluded that the lower the transfer agent concentration, the higher the molecular weight and the higher the consistency of the gel formed.

The PNAlALA homopolymer formed gels regardless of the polymer concentration but they were fragile and breakable. The lack of suitable organic solvent, the need of a H-bond breaking salt or of high temperature in water, and the presence of residual interactions between macromolecules in solutions appeared technical obstacles to using Size Exclusion Chromatography (SEC) for molecular weight determination. Intrinsic viscosity was also problematic because of intermolecular interactions and concentration-dependent gelation. Although dynamic viscosity was not appropriate to measure molecular weights, this technique was selected as a compromise to compare the different copolymers. Data in Table 1 shows a remarkable correlation between the decrease of the concentration in transfer agent and the increase of dynamic viscosity, since 1, 0.25, and 0.1 M in transfer agent corresponded to approximately 1.4, 1.8, and 8 Pa·s, respectively. Based on the molecular weight of the PNAGA_1_ homopolymer (117,000 g/mol), evaluated by viscometry according to the method proposed by Haas [15], a dynamic viscosity of approximately 1.4 Pa·s likely corresponded to molecular weights in the range of 110,000–130,000 g/mol.

### 2.3. Fourrier Transform Infrared

The spectra of the different polymers appeared to be composed of broad absorption zones that included enlarged and overlapping bands, as is usual for acrylic polymers synthesized by free radical polymerization. The spectra looked similar and could hardly be differentiated by this technique, especially when the only structural difference is a proton replaced by a methyl group. In agreement with the structure of the homo and copolymers, the broad zone located between 3650 and 2900 cm^−1^ included free and bonded NH and NH_2_ stretching vibrations above 3000 cm^−1^, and three small peaks on the 3000–2900 cm^−1^ shoulder due to CH, CH_2_, and CH_3_ stretching. Another group of bands was found in the 1690–1500 cm^−1^ region composed of overlapping primary and secondary amide I and II bands followed by CH, CH_2_, and CH_3_ bending deformation bands around 1450 cm^−1^.

### 2.4. Nuclear Magnetic Resonance

Figure 4 shows the ^1^H-NMR spectra of the different copolymers in comparison with corresponding homopolymers. The peaks were assigned according to the formula (Figure 4 right) using the spectra of the homopolymers as references. The composition of the copolymers in NAlALA units (Table 1) was calculated using the ratio 2 × (methyne **2**)/(methylene **d**) resonances areas corresponding to NAGA and NAlALA units, respectively.

Figure 5 shows the variation of the composition of copolymers relative to the composition of monomer feeds. The linearity confirmed the comparable reactivity of the two monomers.

Figure 6 shows the ^13^C-NMR spectra of the copolymers in comparison with those of the homopolymers. The presence of NAlALA-based units is clearly observed at the level of 48 ppm (carbon **2**) and at 18.2 ppm (carbon **1**). 

### 2.5. Circular Dichroism

Figure 7 left shows the CD spectra of PNAlALA at two concentrations in water. As for any unordered aliphatic polyamide chain, the profile resulted from the overlapping of the n–σ* and the π–π* electronic transitions [23]. The copolymers with NAGA exhibited similar profiles that were directly proportional to the content in NAlALA units as shown in Figure 5 right.

This finding shows that NAGA units did not perturb the chiral properties due to NAlALA ones. Therefore, ellipticity was used to assess the composition of copolymers, as it was performed from NMR data. The linearity agreed with the previous conclusion relative to comparable reactivity of the two monomers.

### 2.6. Reversible Gel↔Sol Transitions

PNAGA-*co*-NAlALA copolymers with similar composition and different molar sizes according to transfer agent and dynamic viscosity (COP50_0.1, 0.25, and 1_) or similar molar size and different compositions (COP25_0.25_, 50_0.25_, and 75_0.25_) were selected to investigate the influence of these characteristics on the concentration-dependence of the gel→sol transition (Figure 8). From the various profiles, it was concluded that the three factors were interdependent. The higher the concentration or the higher the content in alanine-based units (comparison of COP25_0.25_, 50_0.25_, and 75_0.25_), the higher the transition temperature was, and the higher the concentration and the higher the molecular weight (comparison of COP50_0.1, 0.25, and 1_), the higher the transition temperature was. 

### 2.7. Release Sustaining

The potential of NAlALA-based polymers relative to drug delivery was estimated using Methylene Blue and Risperidone as models of hydrophilic and hydrophobic drugs, respectively, as performed in the case of a PNAGA homopolymer [21,24]. Figure 9 compares the release profiles observed for Methylene Blue loaded in gels made of a PNAGA homopolymer and a copolymer rich in NAlALA-based units synthesized in the presence of the same concentration in transfer agent, and thus considered as comparable in terms of molecular weight. There was no significant difference between the two release profiles, thus showing that the presence of NAlALA-based units did not affect the diffusion-controlled release typical of PNAGA.

Similarity between release profiles was also observed in the case of the hydrophobic Risperidone in PNAGA and COP50_0.1_ gels for two polymer concentrations, as shown in Figure 10. In contrast, a significant increase of release rate was observed for the COP75_0.25_ copolymer containing a majority of NAlALA-based units, with the initial burst staying rather low (<10%). In all cases, linear release profiles typical of solubility-control were observed. The rate increase observed in the case of the COP75 was tentatively assigned to the effect of the extra methyl groups present in NAlALA-based units on hydrophobicity-related polymer-water interactions that condition the formation of hydrophobic microdomains, where hydrophobic molecules, including drugs, can be accommodated [24,25,26]. In practice, such accommodation corresponds to an increase of dispersed molecules more prone to diffuse in the external aqueous medium than if they were aggregated in dispersed solid particles. The enhancing effect of NAlALA-based units present in the COP75_0.25_ copolymer should have been larger for the PNAlALA homopolymer. However, the poor characteristics of the gel did not allow obtainment of significant release profiles because of weakness related to hydrophobic interactions. This correlation was supported by the fact that a dried PNAlALA gel was very difficult to solubilize again due to the establishment of short-distance macromolecule-macromolecule hydrophobic interactions on top of the amide-related hydrogen bonds. Such short-distance interactions were less energetic in swollen gels but still effective enough to significantly increase the transition temperatures, as shown by the location of the COP75_0.25_ profile in the temperature scale (Figure 8). Sol↔gel transition temperatures high above body temperature were regarded as a possible drawback relative to using thermos-responsive injectable hydrogels in the field of drug delivery.

## 3. Conclusions

This work shows that thermo-responsive PNAGA-*co*-NAlALA copolymers can be synthesized by radical polymerization in aqueous medium, as was the case for PNAGA homopolymers. At 5% *w/v* and slightly above, all the investigated copolymers exhibited UCST-type gelation with molecular weight and concentration-dependences of the gel↔sol transition. The gels were more or less cohesive and consistent depending on the composition that was accessible from the polymerization feed, and from NMR and CD spectra. Some of the gels showed a potential to serve as injectable drug delivery systems under the sol form below 50 °C, as in the case of PNAGAs. The release of the Methylene Blue hydrophilic model was diffusion-controlled, whereas that of Risperidone was linear and thus solubility-controlled. The greater release rate observed for the copolymer rich in NAlALA-base units was tentatively assigned to the associated hydrophobicity known to increase the local solubility of hydrophobic compounds by dissolution in lipophilic microdomains, as observed for many other amphiphilic copolymers. Unfortunately the effect could not be amplified using the PNAlALA homopolymer. The exploitation of COP copolymers rich in NAlALA-base units as injectable hydrogels seems possible provided the sol→gel transition temperatures allow injection under tissue-respecting conditions, as was possible in the case of some PNAGAs.

## 4. Materials and Methods

### 4.1. Chemicals

The l-Alaninamide hydrochloride was purchased from Ark Pharm (Libertyville, IL, USA). Acryloyl chloride, potassium carbonate, 2-propanol, acetone, methanol, diethyl ether, and dichloromethane were supplied by Aldrich (Saint-Quentin, Fallavier, France). Glycinamide hydrochloride was obtained from Acros Organics Fisher Scientific (Illkirch, France). Di-potassium peroxodisulfate was used as received from VWR (Fontenay-sous-Bois, France). The isoosmolar pH = 7.4/320 mOsm phosphate buffer was made by mixing 3.31 g of NaH_2_PO_4_·H_2_O and 33.77 g of Na_2_HPO_4_·7H_2_O in 1 dm^3^ of water. Spectra/Por® membranes were supplied by Roth Sochiel (Lauterbourg, France). The *N*-acryloyl glycinamide was synthesized as reported in our previous study [21].

#### 4.1.1. Synthesis of *N*-Acryloyl l-Alaninamide (NAlALA)

The l-alaninamide hydrochloride (25 g, 0.2 mol) was introduced in a 1 dm^3^ three-necked reactor equipped with a mechanical stirrer and placed in an ice bath. Argon was allowed to gently flow through the reactor. Next, 50 cm^3^ of cool water was added and the mixture was stirred until complete dissolution. Then, 300 cm^3^ of cool diethyl ether and 140 cm^3^ of cool 2 M (mol/dm^3^ potassium carbonate were introduced successively, prior to dropwise addition of a solution of 20 g (0.22 mol) of freshly distilled acryloyl chloride in 100 cm^3^ of diethyl ether. The mixture was stirred at 0 °C during the addition. At the end, stirring was maintained for 1 h at room temperature. The pH of the medium was lowered to 2 by addition of 6 N ([H^+^]/dm^3^) HCl. The organic layer was removed and the aqueous medium was washed three times with 200 cm^3^ of diethylether. The aqueous solution was discolored with charcoal and filtered on Celite®. The remaining trace of diethylether was eliminated under dynamic vacuum at 20 °C and the pH was returned to neutral with 2 N ([OH^−^/dm^3^] NaOH. Finally the mixture was freeze-dried to yield 66 g of a mixture of NAlALA and potassium chloride. The mixture was washed three times with 400 cm^3^ of a 4/1 *v/v* ethanol: methanol mixture to extract the organic component. The solution was evaporated to yield 28 g of crude NAlALA. After recrystallization in 350 cm^3^ of a 4/1 *v/v* ethanol/acetonitrile mixture, pure NAlALA (17 g) was recovered. This monomer melted at 147.5 °C (yield 62% with respect to the initial amount of alaninamide).

#### 4.1.2. Synthesis of Poly(NAGA-*co*-NAlALA) Copolymers

Typically, a suitable amount of isopropanol was introduced in a 250 cm^3^ flask equipped with a magnetic stirrer together with suitable amount(s) of monomer(s) (0.5 mol/dm^3^ in total) and 10.6 mg of potassium persulfate dissolved in 60 cm^3^ of deionized water. The mixture was degassed using bubbling of argon for 1 h. The flask was then placed in an oil bath at 60 °C and the content was allowed to polymerize for about 48 h. At the end, the viscous mixture was diluted with 100 cm^3^ of hot water to obtain a diluted solution at 50 °C and introduced in a dialysis tube (6/8000 Dalton cut-off) maintained at the same temperature for 48 h, with outer water changed several times up to the absence of diffused material. At the end, the content of the tube was recovered and freeze-dried to yield the polymer.

### 4.2. Methods

#### 4.2.1. Spectral Characterizations

ATR-FTIR spectra were recorded using a FTIR spectrometer Perkin-Elmer Spectrum 100 (Villebon-sur-Yvette, France). Polymers were ground and the powder was placed on the ATR plate. The ^1^H-NMR and ^13^C-NMR spectra were recorded using a Bruker Avance III HD (Wissembourg, France)—400 MHz with temperature set at 100 °C. Typically, approximately 30 mg of polymer was solubilized in 0.6 mL of hot d6-DMSO. 

#### 4.2.2. Dynamic Viscosity

Dynamic viscosity data were obtained using a 4100M DMA Densimeter™ coupled with a Lovis 2000 ME microviscometer accessory from Anton Paar (Les Ulis, France). The polymers were dissolved in hot saline at a concentration of 10 g/dm^3^ and measurements were performed at 20 °C. 

#### 4.2.3. Thermal Characterizations

The gel→sol transition temperature of the various polymer-water systems was determined according to the reverse test tube method. Typically, a suitable amount of dried polymer was introduced in a 4 cm^3^ glass tube together with deionized water or saline to make 1 cm^3^ of mixture that was placed in a water bath at 80 °C up to total dissolution, generally observed after approximately 2 min. The solution was then cooled down to room temperature to check gel formation. The gel was turned back to solution by heating at 80 °C for 5 min and cooled down to 0 °C at 40 °C/min, left at 0 °C for 30 min, and then heated slowly up to 80 °C at 4 °C/min. A first gel→sol transition temperature was determined when the gel started to flow along the tube wall. This cycle was repeated 3 times to yield four successive gel→sol temperatures. A ±2 °C reproducibility of the transition temperature was generally observed after the second cycle. For all the hot polymer solutions forming gels on cooling toward room temperature, the sol→gel transition temperatures were systematically lower but did not differ by more than 3–4 °C from corresponding gel→sol ones.

#### 4.2.4. Chiroptical Characterization

The lALA-related optical activity of homo PNAlALA and of the copolymers was determined by circular dichroism (CD) using a Jasco J810 spectrograph (Lisses, France) equipped with an air cooled 150 W Xenon source. Measurements were performed at 20 °C under the following conditions (*C*_polymer_ = 0.025 and 0.0125 g/dm^3^, 0.1 mm cell path-length). The CD apparatus was purged with nitrogen to remove oxygen and ozone. The cell temperature was controlled by Peltier effect. Data were expressed in term of measured ellipticity.

#### 4.2.5. Formulations of Copolymer/Model Drug Hydrogels

Typically, 66 mg of dried copolymer was introduced in a 4 cm^3^ flat-bottom test tube and dissolved in 1 cm^3^ of phosphate buffer heated at approximately 80 °C using a water-bath. About 30 mg of Risperidone (7.3 × 10^−5^ mol) or 1.25 mg of methylene blue (4 × 10^−6^ mol) was then added to the hot solution and the tube containing the mixture was dipped in an iced-bath for fast gelation, a process necessary to keep the sparingly soluble Risperidone as homogeneous dispersion. 

#### 4.2.6. Release Profiles 

Typically, a tube containing the formulation of interest was placed in a 1 dm^3^ plastic bottle containing 600 cm^3^ of receiving medium. The container was attached to the oscillating plateau of a Heidolph 1000 incubator thermostated at 37 °C. Slow stirring was maintained during the whole release duration. At each selected time point, 1 cm^3^ aliquot was withdrawn that was not replaced by fresh medium, the volume of the receiving medium being large enough to maintain pseudo sink conditions. All the sols kept their volume after gelation, except for the 25/75 gel. The initial volume of this gel showed slow contraction during data collection. When the gel lost adherence to the glass tube, data collection was stopped. All data were expressed as mean values from duplicated or triplicated experiments. 

## Figures and Tables

**Figure 1 gels-05-00013-f001:**
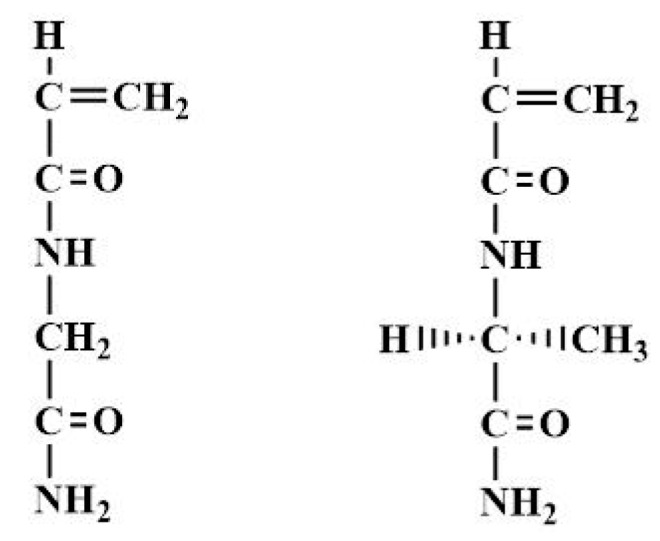
Chemical formula of *N*-acryloyl glycinamide (NAGA) (**left**) and *N*-acryloyl l-alaninamide chiral analog (NAlALA) (**right**).

**Figure 2 gels-05-00013-f002:**
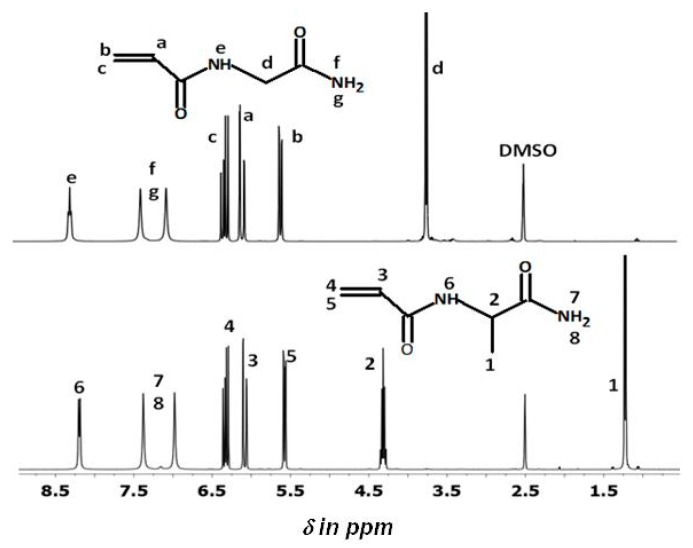
Comparison of the 400 MHz ^1^H-NMR spectra of NAGA (**top**) and NAlALA (**bottom**) in d6-DMSO.

**Figure 3 gels-05-00013-f003:**
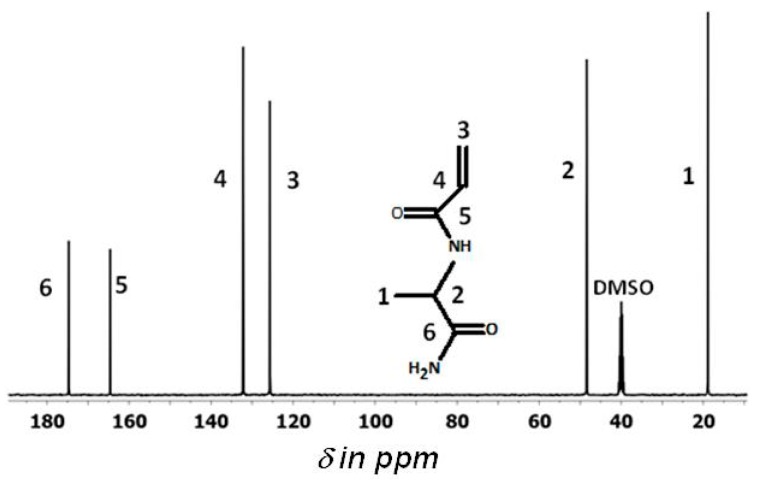
The ^13^C-NMR (400 MHz proton) spectrum of NAlALA in d6-DMSO.

**Figure 4 gels-05-00013-f004:**
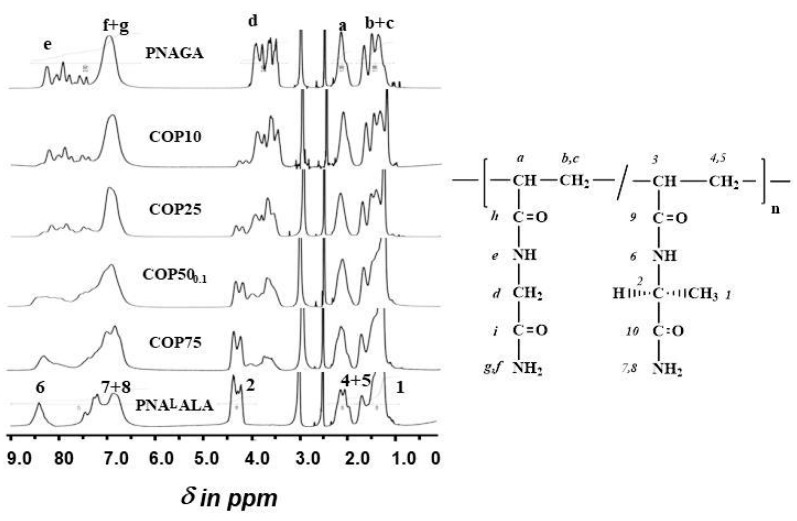
The ^1^H-NMR spectra of homopolymers and copolymers in d6-DMSO at 100 °C (left), and assignments of the resonances to the different protons present in NAGA units (numbers) and in NAlALA ones (letters) (right).

**Figure 5 gels-05-00013-f005:**
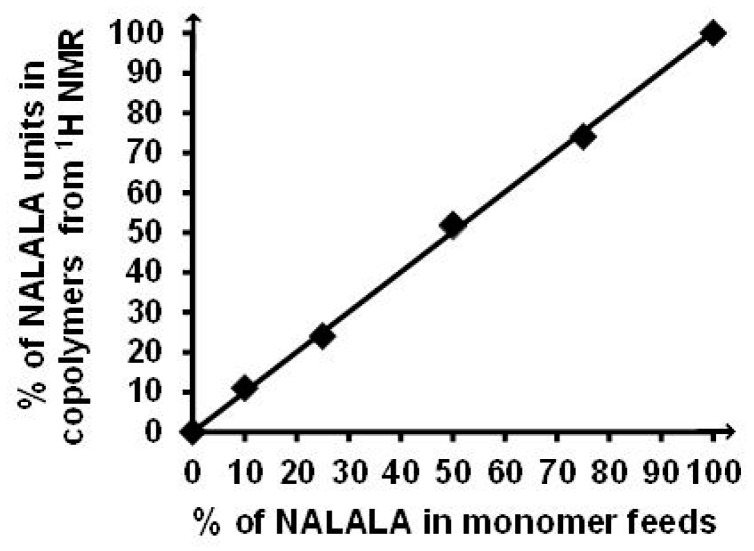
Linear correlation between copolymer and feed compositions. Differences between replicates are inside the size of data points.

**Figure 6 gels-05-00013-f006:**
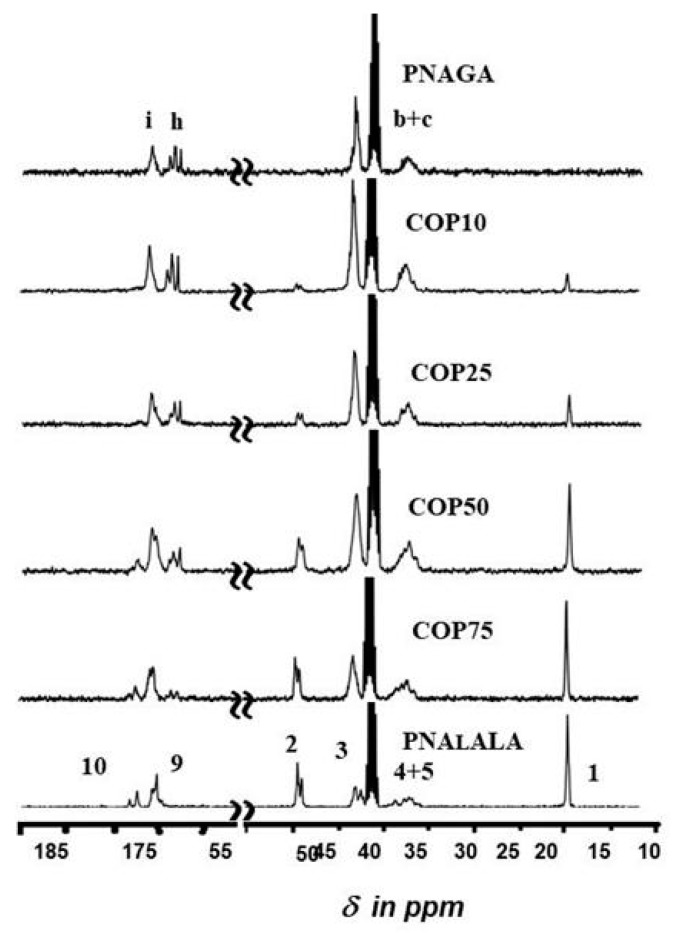
The ^13^C-NMR spectra (**bottom**) of homopolymers and copolymers in DMSO at 100 °C. Assignments of the various carbon atom resonances are based on the formula in Figure 4 (**right**).

**Figure 7 gels-05-00013-f007:**
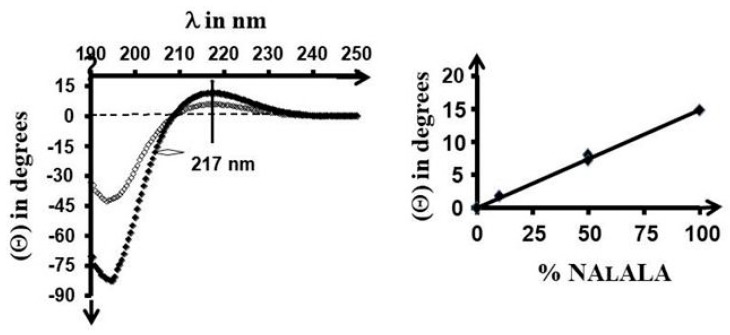
Circular Dichoïsm (CD) spectra of PNAlALA ■: [C] = 0.25% W/V; □: [C] = 0.125% *w/v* in water at 20 °C (**left**) and changes of ellipticity (θ) at 217 nm (**right**) as a function of the composition of copolymers in NAlALA-based units deduced from the ^1^H-NMR spectra shown in Figure 3. Differences in ellipticity of replicates are inside the size of data points, as shown by duplicates for 50% NAlALA.

**Figure 8 gels-05-00013-f008:**
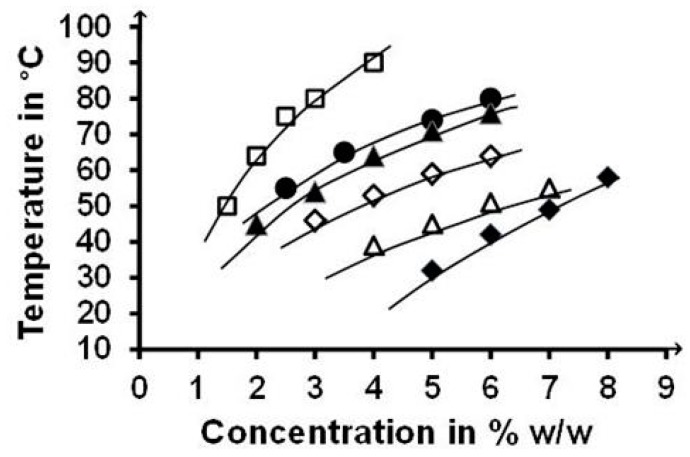
Gel→sol transition temperatures of PNAGA-*co*-NAlALA of different compositions synthesized using different concentrations of transfer agent in the feed: □ Cop75_0.25_; ● Cop50_0.1_; ▲ Cop50_0.25_; ♦ Cop50_1_; ◊ Cop25_0.25_; **∆** Cop10_1_. Differences between replicates are inside the size of data points.

**Figure 9 gels-05-00013-f009:**
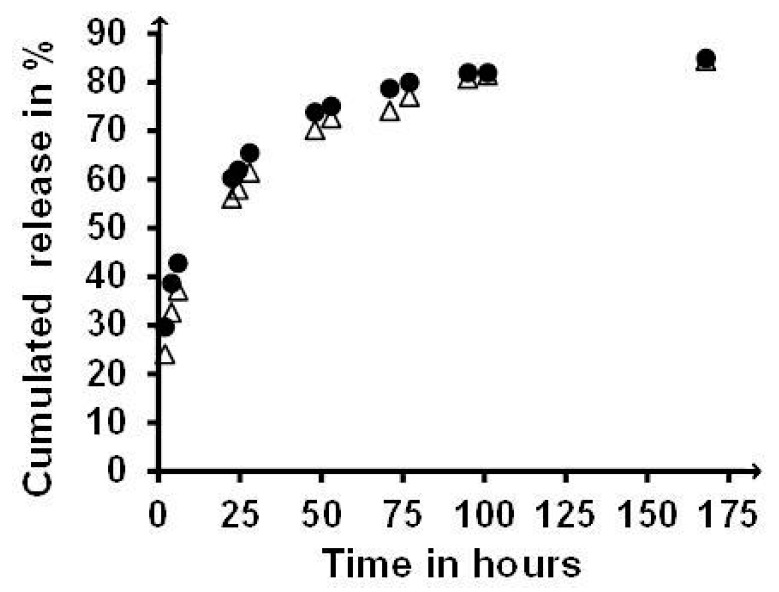
Comparison of the cumulated release profiles of the methylene released from COP50_0.1_ (∆) and PNAGA_0.1_ (●) (gels from 6% polymer solutions) at 37 °C as determined by Ultraviolet (UV) spectroscopy at 665 nm (see the experimental section).

**Figure 10 gels-05-00013-f010:**
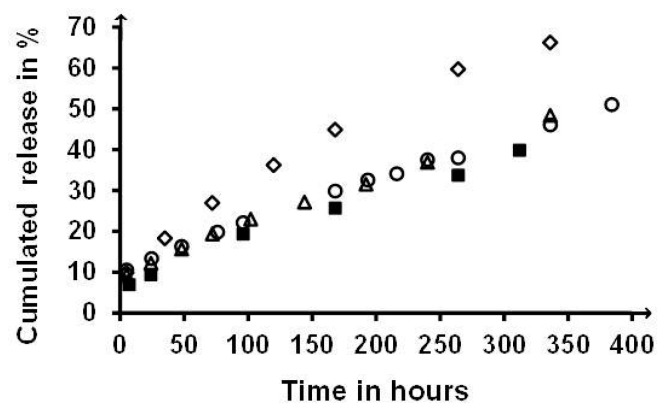
Comparison of Risperidone release profiles from different hydrogels (1 cm^3^ containing 30 mg of Risperidone) in gently stirred 600 cm^3^ of iso-osmolar phosphate buffer at pH = 7.4 and 37 °C: ■: PNAGA_1_ at 6% W/V; ∆ COP50_0.1_ at 6% W/V; ○ COP50_0.1_ at 5 % W/V, and ◊ COP75_0.25_ at 5% W/V. Profiles were averaged from duplicates, and from triplicate for the COP75_0.25_. Differences were all smaller than the data point symbols.

**Table 1 gels-05-00013-t001:** PNAGA, PNAGA-*co*-NAlALA, and PNAlALA polymers obtained by radical polymerization in water at 60 °C initiated by 5 × 10^−4^ M K_2_S_2_O_8_ in the presence of different amounts of isopropanol transfer agent.

Polymer	NAlALA in the Monomer Feed *W*/*W* %		Isopropanol mol/dm^3^	Yield %	Behavior in Saline at Room Temperature		
					Medium Aspect ^2^		Dynamic Viscosity Pa·s
	Feed	Polymer ^1^			1% *W*/*V* ^3^	5% *W*/*V* ^3^	1% *W*/*V* ^3^
PNAGA_1_	0	0	1	93	0	5	1.401 ^4^
COP10_1_	10	9 ± 1	1	95	0	5	1.411
COP25_0.25_	25	22 ± 2	0.25	93	0	6	1.778
COP50_0.1_	50	53 ± 3	0.1	95	2	6	8.534
COP50_0.25_	50	-	0.25	96	0	5	1.810
COP50_1_	50	53 ± 3	1	93	0	2	1.322
COP75_0.25_	75	73 ± 5	0.25	96	0	4	1.918
PNAlALA_1_	100	100	1	90	1	3	n.a.

Note: ^1^ From ^1^H-NMR; ^2^ Physical form: 0: clear solution; 1: solution slightly turbid; 2: clear flowing soft gel; 3: turbid soft and breakable gel; 4: turbid cohesive soft gel; 5: clear breaking gel; 6: elastic gel; ^3^ Concentration in polymer; ^4^ Molecular weight of 117,000 according to the method proposed by Haas [15].

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
