# Peer review of "Poly[(N-acryloyl glycinamide)-co-(N-acryloyl l-alaninamide)] and Their Ability to Form Thermo-Responsive Hydrogels for Sustained Drug Delivery"

_gels, 2019, doi:10.3390/gels5010013_

Reviewer 1 Report

“Poly[(N-acryloyl glycinamide)-co-(N-acryloyl L alaninamide)] and their Ability to Form Thermoresponsive Hydrogels for Sustained Drug Delivery” by Boustta and Vert

This manuscript deals with the formation of (N-acryloyl glycinamide)-co-(N-acryloyl L alaninamide) copolymers, which exhibit an upper critical solution temperature (UCST). The copolymers are capable of forming hydrogels below the UCST and these are explored for potential drug delivery applications. In order to test drug release the authors incorporated methylene blue as a model hydrophilic molecule, and risperidone as a model hydrophobic molecule. The manuscript content is within the scope of Gels and it could be publishable after some revisions.

-          The abstract mentions factors determining transition temperatures such as concentration, average molar mass and copolymerization. The most important factor overlooked by the authors is composition or hydrophilic/hydrophobic character.

-          Acronyms should be defined before they are used in the manuscript (e.g. NMR is used in the abstract instead of Nuclear Magnetic Resonance). The abstract also mentions “Infrared” being used to characterize the polymer. I believe the author intended to write Infrared SPECTROSCOPY or better yet, Fourier Transform Infrared Spectroscopy (FTIR).

-          Mention is made to isoosmolar PBS (another acronym) buffer. If the term isoosmolar is used there should be a reference to a system having the same osmolality. It is likely that the osmolality being referred to is physiological osmolality but this should be stated along with the value for osmolality.

-          Page 1, line 24: “It was found that copolymerization increased the gel→sol transition….” Is this referring to copolymerization with N-acryloyl-alaninamide? Please clarify.

-          Page 1, line 42: “From the viewpoint of environmental conditions the parenteral domain of mammalians is particular in the sense that ionic strength, pH and temperature are controlled.” The use of the term parenteral domain is confusing in this sentence. It is not clear what exactly the authors wanted to convey in this sentence.

-          Page 1, lines 43-44: “The physiological values may serve as critical boundaries relative to exploitation of sol↔gel transitions”.  It seems that physiological values serve more as targets rather than boundaries since they do not vary too much due to homeostasis.

-          The term molecular weights rather than molar masses is preferred throughout the manuscript.

-          Page 2, lines 79-88: Most of this information appears to belong to the experimental section rather than the introduction.

-          Figure 2: the caption should indicate that the top figure is NAGA and the bottom figure NALALA. Alternatively, these figures should be labeled as Figures 2a and 2b and this should be indicated in the caption.

-          Page 4, line 130: Please explain why it was difficult to determine molecular weight using size exclusion chromatography.

-          The use of intrinsic viscosity would appear more suitable as an estimate of molecular weight rather than dynamic viscosity. It is recommended that the authors use intrinsic viscosity or provide a reasonable justification of why it is not possible to make these measurements.

-          A plot of intrinsic viscosity vs. chain transfer agent concentration would be more appropriate.

-          The units of dynamic viscosity are Pa*sec and not Pa/s.

-          Using dynamic viscosity to estimate a range of molecular weights can be misleading. Instead, the intrinsic viscosity should be used.

-          Section 2.3. Are the authors referring to Fourier Transform Infrared Spectroscopy (FTIR)? If so, the IRTF acronym is incorrect. Furthermore, it is better to use the entire name of the technique as a subtitle for this section rather than an acronym.

-          The band assignments in FTIR are not “convincing” to this reviewers. How did the authors determined, unequivocally, that the broad band between 3150 and 3040 cm^-1 is N-H vibration and not unsaturations and/or water vapor? The band at 2938 cm^-1 is ascribed to C-H but it is more likely to come from a CH2 or CH3 vibration mode. Similar comments can be made for the other assignments. It is recommended that the authors consult with somebody with expertise in the interpretation of FTIR spectra of organic compounds.

-          Section 2.4. Use the entire name of the technique rather than the acronym as subtitle.

-          Figure 4. Label the spectra from top to bottom and indicate which spectrum is wich in the figure caption.

-          Figure 5 is missing error bars.

-          Figure 6. Similar comments as those for Figure 4.

-          Figure 8 lacks error bars.

-          Section 2.5. It appears that the authors refer to polymer compositions in abbreviated form such as COP25_0.1 for example to refer to a copolymer consisting of 25% NALALA and 0.1 mol/dm^3 isopropanol. However, this is not explained in the manuscript. It is recommended that the authors describe clearly and unambiguously, their abbreviations. If the first two numbers indicate % NALALA and the second smaller number represents isopropanol molarity, then the description in subsection 2.5, “……..the higher the content in alanine-based units, the lower the transition profile was…….”, is not correct if by transition profile the authors wanted to say transition temperature. If the first number refers to % of NAGA then the statement might be correct. However, it is strange to derive conclusions for the composition of a monomer that is not indicated in the numbers used in the figure. Furthermore, the experimental design in this figure is incomplete since only for Cop50 are all isopropanol compositions being tested. For Cop 75, Cop 25 and Cop10 only one isopropanol concentration is tested and the composition is not the same in all cases (1 M for Cop10, 0.25 M for Cop25 and 0.25 M for Cop75). The incompleteness of the experimental design needs to be justified or the design completed and included in the revised version of the manuscript.

-          Figure 8 also needs error bars.

-          The caption in figures 9 and 10 need to specify the temperature at which the release profiles are being studied.

-          Figure 10 needs error bars so that the reader can determine if the differences in release kinetics for COP75_0.25 are statistically significant.

-          Since the interpretation of the results requires some knowledge about how the experiments were conducted, it may be better to have the experimental section right before results and discussion instead of at the end of the manuscript.

-          Experimental section. Acryloyl chloride dissolved in diethyl ether is added to an aqueous solution. Given the high reactivity of acyl chlorides in water it seems strange that the authors actually were able to obtain the acrylamide compounds. This could be an error in describing the experimental procedure. Other papers in the literature dealing with synthesis of acrylamides or acrylates based on the reaction of acryloyl chloride with amines or hydroxyl groups require extreme caution to remove even trace ammoounts of water. Some even resorting to azeotropic distillations to remove water from reactants. This require an extensive explanation or a correction in the description of the experimental procedures. Interestingly enough, there is no description of thin layer chromatography results when analyzing the reacting mixture and products.

-          Page 9, lines 279-280: “At the end, the viscous mixture was diluted with 100 cm^3  of hot water and introduced in a dialysis tube”. The temperature of water (or temperature ranges) need to be specified. Just mentioning “hot water” is not enough to allow other researchers to duplicate the results.

Author Response

The abstract mentions factors determining transition temperatures such as concentration, average molar mass and copolymerization. The most important factor overlooked by the authors is composition or hydrophilic/hydrophobic character.

The statement the reviewer is referring too is associated with poly(N-acryloyl glycinamide) homopolymers and not with the investigated copolymers. Therefore, we believe that mentioning composition and hydrophilic/hydrophobic character is irrelevant at this stage. No change.

-          Acronyms should be defined before they are used in the manuscript (e.g. NMR is used in the abstract instead of Nuclear Magnetic Resonance). The abstract also mentions “Infrared” being used to characterize the polymer. I believe the author intended to write Infrared SPECTROSCOPY or better yet, Fourier Transform Infrared Spectroscopy (FTIR).

The reviewer is correct. Acronyms have been replaced by full names.

-          Mention is made to isoosmolar PBS (another acronym) buffer. If the term isoosmolar is used there should be a reference to a system having the same osmolality. It is likely that the osmolality being referred to is physiological osmolality but this should be stated along with the value for osmolality.

Isoosmolar PBS was replaced by phosphate buffer saline (PBS). Actually PBS is well defined as a mixture of sodium and potassium chlorides and di-sodium and monopotassium phosphates in amounts suitable to end up with an isoosmolar and isotonic buffer at physiological pH. So isoosmolar PBS was replaced by phosphate buffer (pH=7.4, isotonic to  300 mOsm/dm3). Furthermore, phosphate buffer was added to the materials section. 

-          Page 1, line 24: “It was found that copolymerization increased the gel→sol transition….” Is this referring to copolymerization with N-acryloyl-alaninamide? Please clarify.

The sentence is now : “It was found that increasing the content in N-acryloyl-alaninamide-based units increased the gel→sol transition temperature ……”.

-          Page 1, line 42: “From the viewpoint of environmental conditions the parenteral domain of mammalians is particular in the sense that ionic strength, pH and temperature are controlled.” The use of the term parenteral domain is confusing in this sentence. It is not clear what exactly the authors wanted to convey in this sentence.

The reviewer was right. The text was simplified as : “In mammalians, ionic strength, pH and temperature are set at physiological values that may serve as references to make an in situ sol→gel transition respectful of living elements (proteins, cells, etc.) and exploitable in therapy.”

-          Page 1, lines 43-44: “The physiological values may serve as critical boundaries relative to exploitation of sol↔gel transitions”.  It seems that physiological values serve more as targets rather than boundaries since they do not vary too much due to homeostasis.

The previous correction was complemented as:  For instance, a stimulus-responsive gel-forming system can be injected in the sol form slightly above or below such a physiological reference and gel in situ soon after being under physiological conditions. ..…”

-          The term molecular weights rather than molar masses is preferred throughout the manuscript.

IUPAC proposes both terms alternatively. We followed the reviewer and made the requested change.

-          Page 2, lines 79-88: Most of this information appears to belong to the experimental section rather than the introduction.

The manuscript was constructed according to the instructions to authors and the template provided by the journal. This part was placed at the end of the introduction to help the reader knowing what was done and why in a few words. So we did not change the organization of the paper.

-          Figure 2: the caption should indicate that the top figure is NAGA and the bottom figure NALALA. Alternatively, these figures should be labeled as Figures 2a and 2b and this should be indicated in the caption.

Figure 1 introduced the formula of two monomers. There were used in Figure 2 to identify the spectra. To follow the reviewer, top and bottom were added to the caption. 

-          Page 4, line 130: Please explain why it was difficult to determine molecular weight using size exclusion chromatography.

The text was modified as: “The lack of suitable organic solvent, the need of a H-bond breaking salt or high temperature in water, and the presence of residual interactions between macromolecules in aqueous solutions appeared technical obstacles to using SEC for molecular weight determination. Intrinsic viscosity was also problematic because of concentration-dependent gelation. Although dynamic viscosity was not appropriate to measure molecular weights, this technique was selected as compromise to compare the different copolymers.

-          The use of intrinsic viscosity would appear more suitable as an estimate of molecular weight rather than dynamic viscosity. It is recommended that the authors use intrinsic viscosity or provide a reasonable justification of why it is not possible to make these measurements.

Action included in the answer to the previous comment.

-          A plot of intrinsic viscosity vs. chain transfer agent concentration would be more appropriate.

See the answers to the two previous comments.

-          The units of dynamic viscosity are Pa*sec and not Pa/s.

We thank very much the reviewer. The unit was corrected were the typing was wrong.

-          Using dynamic viscosity to estimate a range of molecular weights can be misleading. Instead, the intrinsic viscosity should be used.

As mentioned before, this technique was used as a compromise and data were compared to the known molecular weight of a member of the family.

-          Section 2.3. Are the authors referring to Fourier Transform Infrared Spectroscopy (FTIR)? If so, the IRTF acronym is incorrect. Furthermore, it is better to use the entire name of the technique as a subtitle for this section rather than an acronym.

Correct. The acronym was turned to its English form whenever it was necessary and full name was used in the sub-title.

-          The band assignments in FTIR are not “convincing” to this reviewers. How did the authors determined, unequivocally, that the broad band between 3150 and 3040 cm^-1 is N-H vibration and not unsaturations and/or water vapor? The band at 2938 cm^-1 is ascribed to C-H but it is more likely to come from a CH2 or CH3 vibration mode. Similar comments can be made for the other assignments. It is recommended that the authors consult with somebody with expertise in the interpretation of FTIR spectra of organic compounds.

We thank the reviewer very much because he was pertinent in pointing out inconsistencies, especially regarding the presence of unsaturation that could not be present in our polymers. We are very sorry. The mistake came from a wrong pasting (the text prepared for the monomers) pasted in the polymer section. This is now corrected as : The spectra of the different polymers appeared composed of broad absorption zones that included  enlarged and overlapping bands as usual for acrylic polymers synthesized by free radical polymerization. The spectra looked similar and could hardly be differentiated by this technique, especially when the only difference is a proton replaced by a methyl group. In agreement with the structure of the homo and copolymers, the broad zone located between 3650 and 2900 cm-1 included free and bonded NH and NH2 stretching vibrations above 3000 cm-1, and three small peaks on the 3000-2900 cm-1 shoulder due to CH, CH2, CH3 stretching. Another group of bands was found in the 1690-1500 cm-1 region composed of overlapping primary and secondary amide I and II bands followed by CH, CH2  and CH3 bending deformation bands around 1450 cm-1.

-          Section 2.4. Use the entire name of the technique rather than the acronym as subtitle.

The recommendation was retained. Correction was done accordingly.

-          Figure 4. Label the spectra from top to bottom and indicate which spectrum is wich in the figure caption.

We decided that indicating the components of the pile of changing spectra directly on the figure was an easier means to identifying and comparing them visually than labelling the spectra in one way or another and then looking to the caption. So nothing was changed.

-          Figure 5, 6, 8 and 10 are missing error bars.

The reviewer was correct. We did not indicate error bars because differences between replicates were so small that they were always inside the size of visible averaged data points for reasonably sized figures. This fact was added to captions.

-          Section 2.5. It appears that the authors refer to polymer compositions in abbreviated form such as COP25_0.1 for example to refer to a copolymer consisting of 25% NALALA and 0.1 mol/dm^3 isopropanol. However, this is not explained in the manuscript. It is recommended that the authors describe clearly and unambiguously, their abbreviations. If the first two numbers indicate % NALALA and the second smaller number represents isopropanol molarity, then the description in subsection 2.5, “……..the higher the content in alanine-based units, the lower the transition profile was…….”, is not correct if by transition profile the authors wanted to say transition temperature. If the first number refers to % of NAGA then the statement might be correct. However, it is strange to derive conclusions for the composition of a monomer that is not indicated in the numbers used in the figure. Furthermore, the experimental design in this figure is incomplete since only for Cop50 are all isopropanol compositions being tested. For Cop 75, Cop 25 and Cop10 only one isopropanol concentration is tested and the composition is not the same in all cases (1 M for Cop10, 0.25 M for Cop25 and 0.25 M for Cop75). The incompleteness of the experimental design needs to be justified or the design completed and included in the revised version of the manuscript.

The construction of abbreviations have been introduced in the text to make Table 1 clearer.

The comparison between COP copolymers relative to molecular weights and composition was also made clearer (see above).

The reviewer pointed out a mistake left after reversing the statement partially only on drafting. The text was corrected as :…. the higher the content in alanine-based units (COP250.25, 500.25, and 750.25),, the higher the transition temperature  was … , and : the higher the concentration and the higher the molecular  weight (COP500.1, 0.25, and 1), the higher the transition temperature was.

-          The caption in figures 9 and 10 need to specify the temperature at which the release profiles are being studied.

The temperature (37°C) was indicated in the caption of Figure 9 but not in that of Figure 10. The temperature was added to the revised caption of Figure 10.

-          Figure 10 needs error bars so that the reader can determine if the differences in release kinetics for COP75_0.25 are statistically significant.

See above

-          Since the interpretation of the results requires some knowledge about how the experiments were conducted, it may be better to have the experimental section right before results and discussion instead of at the end of the manuscript.

We followed the template and by experience, reading the experimental details first does not prevent from returning to the experimental section when getting to data and discussion wherever the experimental section is located. So we kept the format of the template. 

-          Experimental section. Acryloyl chloride dissolved in diethyl ether is added to an aqueous solution. Given the high reactivity of acyl chlorides in water it seems strange that the authors actually were able to obtain the acrylamide compounds. This could be an error in describing the experimental procedure. Other papers in the literature dealing with synthesis of acrylamides or acrylates based on the reaction of acryloyl chloride with amines or hydroxyl groups require extreme caution to remove even trace amounts of water. Some even resorting to azeotropic distillations to remove water from reactants. This require an extensive explanation or a correction in the description of the experimental procedures.

The method used is the classical interfacial reaction between a hydrochloride d’amine located in the aqueous phase with an acyl chloride in the organic phase, in the present case diethyl ether. There is no error here and the protocol is clearly described and conformed to the relevant literature. Nothing changed.

Interestingly enough, there is no description of thin layer chromatography results when analyzing the reacting mixture and products.

We are very sorry but we did not catch the interest of TLC analysis at the level of the reaction mixture. On the other hand, the recovered monomer was purified by recrystallization and it showed sharp melting peak by DSC. Furthermore, the copolymerization, for which purity is important, proceeded nicely. So, TLC was not considered as necessary.

-          Page 9, lines 279-280: “At the end, the viscous mixture was diluted with 100 cm^3  of hot water and introduced in a dialysis tube”. The temperature of water (or temperature ranges) need to be specified. Just mentioning “hot water” is not enough to allow other researchers to duplicate the results.

The text was slightly modified to introduce the temperature and precise the protocol.

Reviewer 2 Report

This is a very nice study. The results are correctly described and discussed.

The major finding of this study is that NaLALA units bring some hydrophobicity in the system. A more precise investigation about the presence of the suspected hydrophobic domains could be performed by using pyrene fluorescence as a probe.

Author Response

We thank the reviewer very much for his favorable opinion and good perception of our work. There was no action done regarding the use of pyrene to investigate deeply the generated hydrophoby. Given the number of polymers to be considered, this could hardly be included in the present work but we may be forced to make such investigation in the future using Risperidone preferably because it is the compound of interest whereas data with pyrene would be more academic and probably difficult to exploit at the level of Risperidone whose physicochemical characteristics are different.

Reviewer 3 Report

The paper  of Boustta et al reports the synthesis of N-acryloyl glycinamide/N-acryloyl L-alaninamide statistical copolymers P(NAGA-co-NALALA) in various compositions by free radical polymerization. These copolymers where characterized by various techniques and their gel-sol transitions, generated upon heating (UCST type) were investigated as a function of polymer concentration. Finally hydrogels of this type were tested as carriers of hydrophilic and hydrophobic drugs. The results seems interesting and might be published. However, in this study some important characteristics, dealing with these novel copolymers are missing. For instance the central property of interest is the UCST of the copolymers and how it varies with the monomer composition. Cloud points should be provided. Moreover, molecular weights for novel polymers should be provided too. The determination of the dynamic viscosity of these associative polymers (due to H-bonding) in low temperature (20 oC) cannot be directly correlated to the molecular weight. The very simple technique used to determine the gel-sol transition is not accurate. Rheology should be involved, which allows to observe phenomena hysteresis between heating cooling cycles. To my opinion, the paper is not suitable for publication as it is.

Author Response

The paper  of Boustta et al reports the synthesis of N-acryloyl glycinamide/N-acryloyl L-alaninamide statistical copolymers P(NAGA-co-NALALA) in various compositions by free radical polymerization. These copolymers where characterized by various techniques and their gel-sol transitions, generated upon heating (UCST type) were investigated as a function of polymer concentration. Finally hydrogels of this type were tested as carriers of hydrophilic and hydrophobic drugs. The results seem interesting and might be published. However, in this study some important characteristics, dealing with these novel copolymers are missing. For instance the central property of interest is the UCST of the copolymers and how it varies with the monomer composition. Cloud points should be provided.

We knew that cloud points are central to most of the literature in which PNAGA polymers are involved. However, cloud points are observed for water-PNAGA or water-PNAGA-copolymers systems in which the polymer concentrations are much lower than those necessary to lead to gelation. Our goal was comparing the copolymers as source of more or less exploitable hydrogels. So determining cloud points is far from being central although it could be so for academic purposes. In order to avoid dilution of our work, the understandable suggestion of the reviewer was not retained.

 Moreover, molecular weights for novel polymers should be provided too. The determination of the dynamic viscosity of these associative polymers (due to H-bonding) in low temperature (20oC) cannot be directly correlated to the molecular weight.

As we faced problems to determined molecular weights, we compromised using dynamic  viscosity. The text has been complemented in order to indicate the problems clearly. The reviewer is right when indicating that dynamic viscosity at 20°C cannot be directly correlated to molecular weights. This is the reason why we used an estimation relative to a PNAGA exhibiting similar dynamic viscosity. This was emphasized in the text in connection with a remark on dynamic viscosity made by reviewer 1.   

The very simple technique used to determine the gel-sol transition is not accurate. Rheology should be involved, which allows to observe phenomena hysteresis between heating cooling cycles.

The reverse tube technique is definitely simple and currently used in the literature of thermo-responsive hydrogels. We used this simple technique to go from gel to sol because, on the one hand,  the sol form is an important phenomenon to allow injection in a warm sol form at a temperature not too much above body temperature, this to fulfil the biocompatibility criterion. On the other hand, gelation must occur rapidly below 37°C after injection as discussed. Although deep investigation of the hysteresis was not important for the main goal of this work, the difference between the gelsol and the sol→gel (stop of flowing) transition temperatures (positive ramp and negative ramp respectively) was rather small (no more than 2°C) for the copolymers systems showing gelation. As we could hardly follow the reviewer and report on hysteresis characteristics for all the copolymers and all the concentrations considered in Figure 8, we decided to limit our reaction to the request by mentioning the small difference in a new sentence in the experimental part. 

 To my opinion, the paper is not suitable for publication as it is.

As indicated, we believe that the reviewers’ comments are correct from an academic viewpoint but the requests are not really necessary when the context of our work is taken into account.

Round  2

Reviewer 1 Report

For the most part, the authors have addressed my review. Just a few comments (my latest comments are underlined):

The abstract mentions factors determining transition temperatures such as concentration, average molar mass and copolymerization. The most important factor overlooked by the authors is composition or hydrophilic/hydrophobic character.

The statement the reviewer is referring too is associated with poly(N-acryloyl glycinamide) homopolymers and not with the investigated copolymers. Therefore, we believe that mentioning composition and hydrophilic/hydrophobic character is irrelevant at this stage. No change.

If the authors were referring to homopolymers, what is the relevance of mentioning copolymerization then? It is one or the other. I recommend reviewing this inconsistency.

- Mention is made to isoosmolar PBS (another acronym) buffer. If the term isoosmolar is used there should be a reference to a system having the same osmolality. It is likely that the osmolality being referred to is physiological osmolality but this should be stated along with the value for osmolality.

Isoosmolar PBS was replaced by phosphate buffer saline (PBS). Actually PBS is well defined as a mixture of sodium and potassium chlorides and di-sodium and monopotassium phosphates in amounts suitable to end up with an isoosmolar and isotonic buffer at physiological pH. So isoosmolar PBS was replaced by phosphate buffer (pH=7.4, isotonic to 300 mOsm/dm3). Furthermore, phosphate buffer was added to the materials section.

The reply that PBS is well defined is not correct. There are many different types of PBS, one of which is Dulbecco’s. But others exist with or without Ca++ or with Mg++, etc. This is why it is important to define the solution so others can reproduce the results.

- Figure 4. Label the spectra from top to bottom and indicate which spectrum is wich in the figure caption.

We decided that indicating the components of the pile of changing spectra directly on the figure was an easier means to identifying and comparing them visually than labelling the spectra in one way or another and then looking to the caption. So nothing was changed.

This is correct. The spectra were already labeled

Interestingly enough, there is no description of thin layer chromatography results when analyzing the reacting mixture and products.

We are very sorry but we did not catch the interest of TLC analysis at the level of the reaction mixture. On the other hand, the recovered monomer was purified by recrystallization and it showed sharp melting peak by DSC. Furthermore, the copolymerization, for which purity is important, proceeded nicely. So, TLC was not considered as necessary.

But it would have given readers an idea of the yields expected if they  wanted to reproduce the results.

Author Response

The abstract mentions factors determining transition temperatures such as concentration, average molar mass and copolymerization. The most important factor overlooked by the authors is composition or hydrophilic/hydrophobic character.

The statement the reviewer is referring to is associated with poly(N-acryloyl glycinamide) homopolymers and not with the investigated copolymers. Therefore, we believe that mentioning composition and hydrophilic/hydrophobic character is irrelevant at this stage. No change.

If the authors were referring to homopolymers, what is the relevance of mentioning copolymerization then? It is one or the other. I recommend reviewing this inconsistency.

Sorry, we misunderstood the point. The sentence was complemented as “…and, in the case of copolymers, composition and hydrophilic/hydrophobic character. “

- Mention is made to isoosmolar PBS (another acronym) buffer. If the term isoosmolar is used there should be a reference to a system having the same osmolality. It is likely that the osmolality being referred to is physiological osmolality but this should be stated along with the value for osmolality.

Isoosmolar PBS was replaced by phosphate buffer saline (PBS). Actually PBS is well defined as a mixture of sodium and potassium chlorides and di-sodium and monopotassium phosphates in amounts suitable to end up with an isoosmolar and isotonic buffer at physiological pH. So isoosmolar PBS was replaced by phosphate buffer (pH=7.4, isotonic to 300 mOsm/dm3). Furthermore, phosphate buffer was added to the materials section.

The reply that PBS is well defined is not correct. There are many different types of PBS, one of which is Dulbecco’s. But others exist with or without Ca++ or with Mg++, etc. This is why it is important to define the solution so others can reproduce the results.

Here again our answer was confusing, since in fact PBS was replaced by phosphate buffer in the text. To make the point clear, the composition of the used buffer was given precisely in the experimental section as requested by the reviewer.

- Figure 4. Label the spectra from top to bottom and indicate which spectrum is wich in the figure caption.

We decided that indicating the components of the pile of changing spectra directly on the figure was an easier means to identifying and comparing them visually than labelling the spectra in one way or another and then looking to the caption. So nothing was changed.

This is correct. The spectra were already labeled

- Interestingly enough, there is no description of thin layer chromatography results when analyzing the reacting mixture and products.

We are very sorry but we did not catch the interest of TLC analysis at the level of the reaction mixture. On the other hand, the recovered monomer was purified by recrystallization and it showed sharp melting peak by DSC. Furthermore, the copolymerization, for which purity is important, proceeded nicely. So, TLC was not considered as necessary.

But it would have given readers an idea of the yields expected if they  wanted to reproduce the results.

We are sorry but we only know TLC as means to verify purity (provided that a suitable mobile phase is found). We do not know how to use this technique to determine yields. This is the reason why a yield determined by weighting was indicated after purification and DSC control at the end of NALALA synthesis, and yields for the different polymers were indicated in Table 1 using the same method after dialysis and drying.

 Reviewer 3 Report

Although  Authors agree with my comments from the academic point of view they argued that it is not necessary to perform the experiments needed to improve the quality of the paper. Anyway the paper could be published in its revised version.  

Author Response

We thank the reviewer very much for his understanding of our strategy.